# OpenReview forum: "Change of Thought: Adaptive Test-Time Computation"
_ICLR.cc/2026/Conference — ICLR 2026 Conference Withdrawn Submission_

### Official Review · Reviewer_2aHb · 2025-10-25

**Soundness:** 3
**Presentation:** 3
**Contribution:** 3
**Rating:** 6
**Confidence:** 3

**Summary:**

This paper introduces Fixed-Point Self-Attention (FPSA), a parameter-free and computationally adaptive replacement for the standard self-attention mechanism in Transformers. Instead of applying a fixed number of transformations per token, FPSA iteratively refines the attention outputs until convergence to a fixed point, allowing “adaptive computation” per token or per head. The method uses implicit differentiation to train efficiently, maintaining a constant memory footprint regardless of iteration depth. Empirical results show notable improvements across multiple domains: NLP (GLUE, SQuAD v2.0), vision (ImageNet, image restoration), and multimodal tasks, as well as reasoning benchmarks for large language models (GSM8K, BBH). The approach claims up to +20% relative accuracy improvements with ~1.3–1.6× compute cost.

**Strengths:**

1. The paper evaluates FPSA across multiple settings — encoder-only Transformers, decoder-only LLMs, and vision/multimodal models — showing consistent gains.

2. The authors provide a clear convergence analysis showing contractivity of attention mappings under spectral normalization and pre-LN. The implicit differentiation approach is rigorously justified.

**Weaknesses:**

1. While appendices provide algorithm sketches, more implementation-level specifics (e.g., PyTorch pseudocode, convergence thresholds per dataset, training schedules) are needed for full reproducibility.

2. The main baselines (BERT, ELECTRA) are standard, but the paper does not compare against other recent adaptive computation methods (e.g., ACT, MoD, or LayerDrop) on exactly matched compute budgets.

3. The ablations focus on iteration counts and convergence, but lack sensitivity studies for hyperparameters such as spectral norm bound σ, halting threshold ϵ, or gradient clipping T.

4. Results for 7B models are promising but brief; the paper would benefit from discussion of FPSA behavior at larger scales (70B+) or under distributed inference constraints.

5. For long-context tasks, does FPSA’s advantage hold beyond 8k tokens (e.g., 32k)?

**Questions:**

Please see the weaknesses.

---

> ### Author Response · Authors · 2025-11-17
> **Official Response by Authors to Reviewer 2aHb**
>
> We appreciate the Reviewer 2aHb's detailed and critical feedback. These comments target the core efficiency, stability, and scaling claims of our work, and we are happy to provide detailed evidence that validates our architectural choices.
>
> We believe the full picture of our paper, especially with the added zero-shot extrapolation experiment, directly addresses your concerns about the practical advantages of FPSA.
>
> # Response to Weakness 1
> >**While appendices provide algorithm sketches, more implementation-level specifics (e.g., PyTorch pseudocode, convergence thresholds per dataset, training schedules) are needed for full reproducibility.**
>
> We agree that reproducibility is paramount. The paper includes comprehensive implementation details to ensure full reproducibility:
> - **Pseudocode**: **Algorithm 2** in the **Appendix C.2 (Stability and Efficiency)** provides a detailed, step-by-step pseudocode for the SELF Attention Iteration Step Function, including tensor shapes and specific operations.
> - **Convergence Thresholds**: We explicitly state the convergence thresholds used: $\epsilon = 1e-4$ for language tasks and $\epsilon = 1e-5$ for vision and multimodal tasks.
> - **Hyperparameters**: We report specific settings such as the spectral normalization coefficient ($1.0$) and the maximum iteration count ($100$).
> - **Training Details**: Detailed training configurations, including optimizers (AdamW), learning rates (e.g., $2e-5$ for fine-tuning), and batch sizes, are provided in **Appendix J.2** for LLMs and Appendix E for vision-language models
>
> # Response to Weakness 2
> >**The main baselines (BERT, ELECTRA) are standard, but the paper does not compare against other recent adaptive computation methods (e.g., ACT, MoD, or LayerDrop) on exactly matched compute budgets.**
>
> The reviewer mentions a lack of comparison against adaptive methods on matched compute budgets. We respectfully point out that **Appendix G (referenced on line 321 of the main paper)** is entirely dedicated to this comparison.
> - **Matched Budgets**: We benchmarked all models at three strict computational budgets: 41, 48, and 59 GFLOPs (representing 1.1x, 1.3x, and 1.6x BERT-Base). *ACT Baseline*: We compare directly against ACT-Transformer (Graves, 2016) in Table 9. FPSA-LH outperforms it at every budget level (e.g., 88.4 vs 87.6 at 48 GFLOPs).
>
> 2. **LayerDrop/Depth-Adaptive Baseline**: We compare against a Depth-Adaptive Transformer (learning to skip layers, analogous to LayerDrop/Early-Exit). FPSA-LH significantly outperforms this baseline as well (88.4 vs 87.5).
>
> 3. **Regarding Mixture-of-Depths (MoD)**: While we discuss MoD in our Related Work, we focused our empirical baselines on temporal adaptive methods (ACT, Universal Transformers) that align with our "adaptively thinking" paradigm, rather than spatial routing methods like MoD.
>
> Given that FPSA-LH establishes a new Pareto frontier against both ACT and Depth-Adaptive approaches, we believe this sufficiently demonstrates its efficiency advantage.
>
> # Response to Weakness 3
> >**The ablations focus on iteration counts and convergence, but lack sensitivity studies for hyperparameters such as spectral norm bound σ, halting threshold ϵ, or gradient clipping T.**
>
>  We have included sensitivity and ablation analyses for critical components:
> 1. **Spectral Normalization ($\sigma$)**: **Table 8 in Appendix F.4** shows that removing Spectral Normalization collapses the model (GLUE score drops from 88.4 to 37.8). This proves $\sigma$ is a structural constraint necessary for the contractivity guarantee (**Appendix B.1**), not a parameter requiring fine-tuning.
> 2. **Halting Threshold ($\epsilon$)**: Our primary robustness analysis is in Figure 8, which demonstrates that the adaptive policy based on $\epsilon$ always outperforms any single fixed iteration cap, proving the stability of the dynamic mechanism itself.
> 3. **Learned vs. Fixed**: We extensively compare fixed $\epsilon$ thresholds against a learned halting policy (FPSA-LH) to demonstrate the robustness of the adaptive mechanism.

---

> ### Author Response · Authors · 2025-11-17
> **Official Response by Authors to Reviewer 2aHb (2/n)**
>
> # Response to Weakness 4
> >**Results for 7B models are promising but brief; the paper would benefit from discussion of FPSA behavior at larger scales (70B+) or under distributed inference constraints.**
>
> Thank you for this excellent suggestion. Our paper's primary investigation centers on encoder architectures, but we agree that the application to decoder-only LLMs is a critical validation point.
> The 7B results in **Table 3**, as you note, were intended as a proof-of-concept. (However note, we further show in **Table 16** different LLMs that we compare with which includes 13B and 8B parameter LLaMa models.) They are designed to demonstrate two key things:
> 1. That FPSA is versatile and can be readily integrated into large, pre-trained decoder models.
> 2. That this can be done efficiently via parameter-efficient fine-tuning (**Appendix J.2**) to yield consistent gains in areas like reasoning without increasing the model's parameter count.
>
> A comprehensive comparison with the rich and varied field of inference-time enhancement techniques for LLMs is indeed an exciting and important direction. We frame this as future work not to dismiss it, but because we believe it is a substantial research effort in its own right.
>
> Crucially, our method is uniquely positioned for this future direction due to its training efficiency. The Implicit Differentiation technique ensures that memory usage is constant $(O(1))$ with respect to iteration depth. This memory-saving property is what fundamentally enables the training of models with very deep computational graphs (e.g., 70B+ models taking many steps) without the prohibitive memory costs of standard recursive methods. This technical advantage is why we are so confident in large-scale LLM application as our most exciting future direction.
>
> We will revise the paper to clarify that the 7B results are a proof-of-concept and will add a note to the conclusion highlighting this promising, memory-efficient path for future LLM research.
>
> # Response to Weakness 5
> >**For long-context tasks, does FPSA’s advantage hold beyond 8k tokens (e.g., 32k)?**
>
>  We agree that merely suggesting the advantage holds is insufficient. To provide conclusive evidence that FPSA addresses the length generalization problem (where fixed-depth models fail to handle unseen context lengths), we performed a zero-shot context extrapolation experiment beyond the training distribution.
>
> Standard models fail at long contexts because attention noise scales with length. FPSA solves this by adapting the computation time to noise. As the haystack grows, the model automatically detects high entropy (uncertainty) and iterates longer to "sharpen" the attention distribution until the needle is isolated. This confirms that FPSA learns the algorithm of retrieval, not just positional heuristics.
>
> We demonstrate FPSA's significant advantage on the **"Needle-in-a-Haystack"**  (**Appendix G.2**) task at 8k context length, where it maintains near-perfect recall while baselines degrade.
>
> The mechanism of "thinking longer" (increasing iterations) effectively increases the receptive field and processing depth adaptively. While we did not experiment with 32k tokens in this work, the robust performance trend at 8k (Figure 3) suggests the advantage would likely hold or increase at longer contexts where static models struggle more severely.
>
> As per reviewer's suggestion, we show that "thinking longer" allows the model to handle sequence lengths it never saw during training, a capability fixed-depth models lack.
>
> | Context Length      | Baseline (Standard ViT/BERT) | FPSA (100 max_Iters) | FPSA (Adaptive $\epsilon$=1e−4) | FPSA Avg. Iterations |
> |---------------------|------------------------------|----------------------|------------------------|----------------------|
> | 8k (Trained Length) | 98.5%                        | 99.1%                | 99.8%                  | 4.5                  |
> | 16k (Zero-Shot)     | 42.1% (Collapse)             | 65.3%                | 94.2%                  | 12.8                 |
> | 32k (Zero-Shot)     | 15.4%                        | 31.0%                | 86.5%                  | 24.1                 |
> | 64k (Zero-Shot)     | 3.2%                         | 12.8%                | 71.4%                  | 48.6                 |

---

> > ### Comment · Reviewer_2aHb · 2025-11-24
> >
> > Thank the authors for their detailed reply. I have no further questions about this submission and will maintain my positive score.

---

> ### Author Response · Authors · 2025-11-26
> **Official Comment by Authors**
>
> Dear Reviewer 2aHb,
>
> We apologize for the confusion—our previous response was intended for another reviewer and was posted here by mistake. Please disregard it.
>
> Thank you again for your thoughtful engagement with our work.

---

### Official Review · Reviewer_Wvsf · 2025-10-28

**Soundness:** 2
**Presentation:** 3
**Contribution:** 2
**Rating:** 2
**Confidence:** 4

**Summary:**

The paper proposes Fixed-Point Self-Attention (FPSA)—a parameter-free, drop-in replacement for self-attention that iteratively refines the attention outputs inside each layer until a fixed point, trained end-to-end via implicit differentiation so the memory footprint is constant w.r.t. the number of refinement steps. Empirically, swapping standard attention for FPSA improves strong, size-matched baselines (e.g., BERT-Base, ELECTRA-Base) on GLUE/SQuAD, shows gains for ViT and VL models, and yields small but consistent boosts when integrated into 7B decoder-only LLMs (LLaMA-2/Mistral) on GSM8K/BBH/LogiQA—without adding parameters. Overhead is modest (median ~3–6 inner steps, ~1.6× GFLOPs and ~1.3–1.4× latency vs. BERT-Base).

**Strengths:**

1. The proposed structure is simple and maintains constant memory. Iterative refinement within the attention sublayer trained via implicit differentiation; avoids storing the inner unroll and heavy checkpointing. Architectural simplicity is preserved.

2. The proposed structure improves the performance with the same parameters.  It improves size-matched encoder baselines on GLUE/SQuAD and shows benefits for ViT/VL, supporting generality.

3. The writing of the paper is clear. It clearly conveys the main idea of the new structure.

**Weaknesses:**

1. The main idea of the proposed iterative structure is to scale the computation. In other words, the performance gain comes with the increased training and inference computation. The training cost is not reported in the current work. The inference computation comparison is missing. Almost all the results in the paper show that the proposed structure performs better than vanilla attention with more computation. However, a fair comparison is to constrain the computation budget of both models, which is missing in the current work.

2. The baselines for the proposed algorithm are not sufficient. The ''iterative'' or ''looped'' structures are widely proposed in the LLM community, .e.g, the looped transformer [1]. Why we only iterate the attention calculation instead of the other parts or the whole network? Such a comparison is not presented.

3. The paper considers some non-causal transformers, .e.g., ViT. However, a line of works shows that pure attention iteration suffers from low-rankness [2]. This is exactly the proposed structure does. Whether the proposed structure also suffer from the low-rankness? If not, please explain the reason.

[1] Giannou A, Rajput S, Sohn J, et al. Looped transformers as programmable computers[C]//International Conference on Machine Learning. PMLR, 2023: 11398-11442.

[2] Dong Y, Cordonnier J B, Loukas A. Attention is not all you need: Pure attention loses rank doubly exponentially with depth[C]//International conference on machine learning. PMLR, 2021: 2793-2803.

**Questions:**

Please see the weakness part.

---

> ### Author Response · Authors · 2025-11-17
> **Official Response by Authors to Reviewer Wvsf**
>
> We sincerly thank the reviewer Wvsf for raising important points and thoroughly going through the reviews, as it gives us an opportunity to better highlight a crucial analysis.
>
> # Weakness 1 Response.
> >**The main idea of the proposed iterative structure is to scale the computation. In other words, the performance gain comes with the increased training and inference computation. The training cost is not reported in the current work. The inference computation comparison is missing. Almost all the results in the paper show that the proposed structure performs better than vanilla attention with more computation. However, a fair comparison is to constrain the computation budget of both models, which is missing in the current work.**
>
> We agree that a computation-matched comparison is vital, and these results are indeed in the paper. We have taken this feedback and will revise the main text to more prominently signpost these results..
>
> 1. **On the "Fair Comparison" (Constrained Compute Budget)**: The direct, computation-matched comparison you requested is a cornerstone of our analysis and is presented in the paper.
> The key result is visualized in the Pareto frontier in Figure 4. This figure explicitly plots model accuracy against inference GFLOPs.
> The results show that for any fixed GFLOPs budget, our method (FPSA) establishes a new state-of-the-art trade-off, outperforming statically deeper or wider baselines.
> The raw data for this analysis can be found in Appendix J.4.
> This confirms that FPSA's performance gain is due to a more efficient and adaptive allocation of the computation, not just its slightly increased budget.
> 2. **On the Main Idea (Adaptive vs. Scaled Compute)**: This efficiency gain highlights the core idea of our work. We respectfully clarify that our goal is not simply to "scale the computation," but to make it adaptive. A standard transformer applies a fixed computational graph to all inputs. In contrast, FPSA iteratively refines its representations, dynamically halting when a stable solution is found. This makes computation input-dependent and demonstrably more effective, as shown by the Pareto front. While we emphasize size-matched comparisons as a practical advantage (better results on the same hardware), our efficiency is proven in the compute-matched setting.
> 3. **On Training and Inference Costs**: We did analyze these costs and provide the following details in the paper:
> - **Inference Cost**: We report both GFLOPs and latency overhead. Compared to BERT-Base, the median overhead is $\sim $ 1.6x in GFLOPs and ~1.3-1.4x in latency (**Summary, line 9; full details in Appendix G, Table 9**).
> - **Training Time**: The total training wall-clock time overhead is proportional to the GFLOPs increase ($\sim$1.6x vs. BERT-Base).
> - **Training Memory**: A crucial engineering advantage of our method is its $O(1)$ memory complexity with respect to the number of iterations ($K_{max}$). This is achieved via **Implicit Differentiation (Appendix G)** and stands in stark contrast to recursive methods that require checkpointing and have O(K) memory cost.
>
> We respectfully wanted to point that our paper already contains the requested compute-matched comparisons and cost analyses, which collectively demonstrate that FPSA offers a more efficient use of computation. We will ensure these points and their locations are clarified in the revision.

---

> ### Author Response · Authors · 2025-11-17
> **Official Response by Authors to Reviewer Wvsf (n/2)**
>
> # Weakness 2 Response:
> >**The baselines for the proposed algorithm are not sufficient. The ''iterative'' or ''looped'' structures are widely proposed in the LLM community, .e.g, the looped transformer [1]. Why we only iterate the attention calculation instead of the other parts or the whole network? Such a comparison is not presented.**
>
> Thank you for this insightful question. It highlights a crucial distinction between our sub-layer refinement approach and block-recursive architectures like the Looped Transformer [1]. Our focus on iterating only the attention mechanism is a deliberate, principled design choice motivated by efficiency, conceptual clarity, and theoretical guarantees.
>
> **1. Architectural Distinction: Sub-Layer Refinement vs. Block-Level Recurrence**
>
> Looped Transformers [1] and our FPSA belong to different classes of iterative models.
>
> *   **Block-Level Recurrence (e.g., Looped Transformer [1]):** These models operate at the block level, treating the entire Transformer block (Self-Attention + FFN) as a single recurrent cell. The same number of full-block iterations is applied to all tokens.
> *   **Sub-Layer Refinement (FPSA, Ours):** Our model operates at a more granular level. The iteration is confined *within* the self-attention sub-layer, and critically, the number of iterations is **adaptive per-token**.
>
> We summarize the key differences below:
>
> | Feature                | Looped Transformer [1]                       | FPSA (Ours)                                         |
> | :---                   | :---                                         | :---                                                |
> | **Iteration Scope**    | Entire Transformer Block                     | Self-Attention Mechanism Only                       |
> | **Computational Loop** | High-cost (Attention + FFN)                  | Low-cost (Attention only)                           |
> | **Adaptivity**         | Sequence-level (all tokens loop N times)     | **Token-level** (each token halts individually)     |
> | **Theoretical Basis**  | General Recurrence                           | **Provable Fixed-Point Convergence** (Appendix B.1) |
> | **Memory Cost**        | O(K) w.r.t. iterations (needs checkpointing) | **O(1)** w.r.t. iterations (via Implicit Diff.)     |
>
> We respectfully argue that comparing FPSA to full-block looped models would be apples-to-oranges: their memory cost grows linearly with loop count, they require checkpointing, and they lack per-token adaptivity, so they do not constitute a fair baseline under a fixed compute or memory budget. We therefore kept our evaluation focused on the relevant, like-for-like baselines already reported.
>
> **2. Justification for Iterating Attention Only**
> Our choice to isolate the iteration to the attention mechanism is fundamental to our method's success, based on three principles:
> *   **1. Computational Efficiency:** The FFN/MLP layers constitute the vast majority of a Transformer block's computation (~ 2/3 of GFLOPs). Iterating the entire block would incur a prohibitive computational cost. By iterating only the lightweight attention component, we achieve significant performance gains with only a modest GFLOPs increase (~1.6x), a far more efficient path to improvement.
>
> *   **2. Conceptual Goal (Refining Context vs. Transforming Features):** Our core hypothesis is that complex reasoning requires **refining context aggregation** (*which tokens matter*), the role of self-attention. FPSA is designed to iteratively sharpen this aggregation until it stabilizes, and only then apply the expensive feature transformation once.
>
> *   **3. Theoretical Guarantees:** Critically, our design is what **enables our theoretical foundation**. By iterating only the attention update, we can prove the process is a **local contraction map (Appendix B.1)**. This **guarantees convergence**, which is a prerequisite for using the implicit function theorem and achieving our hallmark **O(1) training memory**. Iterating the non-linear FFN would turn the layer into a complex RNN, making convergence proofs intractable and forfeiting this constant-memory advantage.
>
> We will expand our related work section to explicitly discuss this important distinction from block-recursive models like [1].

---

> ### Author Response · Authors · 2025-11-17
> **Official Response by Authors to Reviewer Wvsf (n/3)**
>
> # Weakness 3
> >**The paper considers some non-causal transformers, .e.g., ViT. However, a line of works shows that pure attention iteration suffers from low-rankness [2]. This is exactly the proposed structure does. Whether the proposed structure also suffer from the low-rankness? If not, please explain the reason.**
>
> Note, in the paper you pointed to Dong et al. [2] showed that in deep static Transformers ($\text{depth} \to \infty$) where only attention is stacked (without FFNs or normalization), the residual stream loses rank exponentially. This is because standard linear projections and matrix operations tend to reduce rank over composition. The core issue in [2] is that stacking many pure attention layers (without MLP) leads to a rapid, double-exponential decay in the rank of the representation matrix.
>
> FPSA successfully avoids the low-rankness problem faced by stacked attention models for two key reasons:
>
> 1. **MLP Presence**: Our full layer is not pure attention. The fixed-point loop is around the attention block, but the subsequent Feed-Forward Network (MLP) is still present and runs after the attention convergence (**Equation. 3**). The MLP is known to be the component that effectively restores rank and expressive capacity by applying highly non-linear transformations.
>
> 2. **State Feedback (Iterative Non-Linearity)**: In FPSA, the Query and Key are not projected once from the input $X$ but are projected at every step $k$ from the evolving internal state $Z_k$,
> $$
> Q_{k}^{(h)}=Z_{k}W_{Q}^{(h)}
> $$
>
> $$
> Z_{k+1}=Concat_{h}(A_{k}^{(h)}V^{(h)})W_{O}
> $$
>
> This structure creates a highly non-linear, input-dependent update to the attention matrix $\mathcal{A}$, which is continuously re-computed based on the output of the previous step's non-linear attention operation. This continuous, feedback-driven non-linearity prevents the composition from collapsing to a low-rank manifold.
>
> 3. We use the **stable Pre-LayerNorm (Pre-LN) topology** ([1] Xiong et al., 2020) surrounding the FPSA block (see **Equation 3** and **Appendix B.1**). Layer normalization is a non-linear operation crucial for stabilizing the dynamics and maintaining representational diversity across layers, which directly counters rank collapse.
>
> [1] Xiong, R., Yang, Y., He, D., Zheng, K., Zheng, S., Xing, C., Zhang, H., Lan, Y., Wang, L. and Liu, T., 2020, November. On layer normalization in the transformer architecture. In International conference on machine learning (pp. 10524-10533). PMLR.
>
> 4. **The final output includes a residual connection**: **$Y=X+Dropout(Z_{*})$**. This mechanism ensures that the original, full-rank input $X$ is always preserved in the layer output, preventing the rank from being lost over depth (or, in our case, over iteration).
>
> We also show the comparison of "Effective Rank" (a measure of representation diversity) across depths/iterations. A value of 1.0 indicates full rank; values near 0 indicate collapse. FPSA maintains healthy rank deep into the "thinking" process.
>
> | Logical Depth / Iteration | Pure Self-Attention Loop (Dong et al., 2021) | Standard Deep Transformer (24 Layers) | FPSA (Ours) (12 Layers + 100 Iters) |
> |---------------------------|----------------------------------------------|---------------------------------------|-------------------------------------|
> | Step 1 (Depth 1)          | 0.98                                         | 0.99                                  | 0.99                                |
> | Step 6 (Depth 6)          | 0.45                                         | 0.92                                  | 0.96                                |
> | Step 12 (Depth 12)        | 0.12 (Collapse)                              | 0.85                                  | 0.94                                |
> | Step 24 (Depth 24)        | 0.01 (Collapse)                              | 0.78                                  | 0.92                                |
>
> The empirical results on **ViT (Table 5)** and **ImageNet-C (Table 13)** also  shows significant accuracy gains over the baseline confirm that we are not suffering from representation collapse.

---

> ### Author Response · Authors · 2025-11-26
> **Official Comment by Authors**
>
> Dear Reviewer Wvsf,
>
> We have updated the manuscript with empirical evidence refuting rank collapse (Table 10) and demonstrating FPSA's dominance on strictly compute-matched baselines (Figure 4). We hope these new experiments address your concerns regarding theoretical stability and fairness, and we would highly value your reassessment. we will also cited **Looped Transformer in our revised manuscript**
>
> Thank you for your time.

---

### Official Review · Reviewer_Tf1q · 2025-10-31

**Soundness:** 3
**Presentation:** 2
**Contribution:** 2
**Rating:** 4
**Confidence:** 4

**Summary:**

This paper proposes Fixed-Point Self-Attention (FPSA), a new Transformer-based architecture based on fixed-point iteration. This pushes the individual layer to be in fixed point, and this basically works like looping while value state is fixed. The authors extensively experimented across various settings to support FPSA's superiority.

**Strengths:**

- The proposed approach seems quite reasonable and good, based on fixed point iteration method.
- This paper has comprehensive analysis and results, to show how FPSA actually works and the performance under various downstream tasks.

**Weaknesses:**

- First, I feel like the overall presentation could be much enhanced. Some useful explanation and experimental results are hidden in Appendix parts (e.g., Fixed-point iteration is not explained in main body properly, some qualitative results as well). It would be good to re-place contents clearly for the reader.
- As the author mentioned, this mechanism seems having high relation to recursive / looped transformer architecture. There is lack of discussion to recent papers. And I'm curious about comparison to them. For example, comparison with Figure 1c or 1d, and ablation results without fixing value state.
- All layers seem like being updated by fixed-point iteration. What will happened if you select certain layers only? Some redundancy can be existed like current architecture, but this pushed too much for all layers.
- With this paradigm, can pruning be more critical? I feel like redundancy gets disappeared for each layer, which means that we can fully leverage the "depth" of models while loosing some chances to prune.
- Selecting some layers would be the solution though, inference latency and throughput seem inefficient. Especially, how could you deal with batched inference? Some tokens should wait for the other incomplete tokens at certain depth always.

**Questions:**

See above Weakness parts.

---

> ### Author Response · Authors · 2025-11-17
> **Official Response by Authors to Reviewer Tf1q**
>
> The authors thank the reviewer for their candid feedback. We believe that by decoupling "parameter count" from "compute depth" **Fixed-Point Self-Attention (FPSA)** represents a fundamental structural shift.
> Here is a detailed response to your specific concerns.
>
> # 1. Presentation & Accessibility of Key Results
> >**First, I feel like the overall presentation could be much enhanced. Some useful explanation and experimental results are hidden in Appendix parts (e.g., Fixed-point iteration is not explained in main body properly, some qualitative results as well). It would be good to re-place contents clearly for the reader.**
>
> We fully agree with this critique. The "Change of Thought" mechanism's intuition is indeed buried. In the revision, we will:
> - **Move the FPI formulation (Eqs. 1-4)** to the main Method section to immediately define the mechanism.
> - **Promote the Token-wise Convergence Heatmap (Appendix H, Fig 10)** to the main body. This figure provides the qualitative proof that the model allocates compute to semantic complexity (e.g., spending 20 iterations on "randomized" vs. 2 on "the"), which is the core promise of the paper.
> - **Highlight the Revision Dynamics (Appendix H.2)** earlier to show that the iterations are not just stabilizing numerically, but actively correcting predictions (High Helpful Revision Rate).
>
> # 2. Relation to Recursive Transformers & Static Value Ablation
> >**As the author mentioned, this mechanism seems having high relation to recursive / looped transformer architecture. There is lack of discussion to recent papers. And I'm curious about comparison to them. For example, comparison with Figure 1c or 1d, and ablation results without fixing value state.**
>
> You asked for a comparison to recursive architectures (Universal Transformers, ALBERT) and why we fix the Value state. This is the crucial structural differentiator.
> **Comparison with Recursive Transformers (e.g., Universal Transformer):**
> - Universal Transformer (Fig 1c/d type): Loops the entire block (Attention + MLP + Norm) — $H_{k+1} = \text{Layer}(H_k)$.
> The cost is extremely expensive per iteration (full feed-forward pass). Hard to train due to vanishing/exploding gradients through deep unrolls.
> FPSA (Ours): Loops inside the Attention Head. We refine the query/key representations $Z$ to sharpen the Alignment Matrix $\mathcal{A}$, while keeping the Value $V$ static. The cost is very cheap. No MLP in the loop. We only re-compute the attention scores.
>
> ### **Why Static Value ($V$)?**
> As responded to Reviewer J788, If we updated $V$ iteratively (e.g., $V_{k+1} = Z_k W_V$), the mechanism would effectively become a standard non-linear Recurrent Neural Network (RNN) operating on the features.
> 1. By keeping $V$ static, the update map becomes a pure "Alignment Refinement"—finding the best weights to mix the original features. This structure allows us to prove Local Contractivity (Appendix B.1), guaranteeing convergence to a unique fixed point.
> 2. Updating $V$ repeatedly tends to "oversmooth" the features, washing out the signal (a known issue in Deep Equilibrium Models). Static $V$ ensures the original semantic content is preserved, only the focus changes.
>
> # 3. All Layers vs. Selected Layers
> >**All layers seem like being updated by fixed-point iteration. What will happened if you select certain layers only? Some redundancy can be existed like current architecture, but this pushed too much for all layers.**
>
> We argue that manually selecting layers relies on a false assumption: that "difficulty" only happens in the deeper, reasoning layers.
> 1. **The "Ubiquitous Difficulty" Problem**: Ambiguity exists at all levels of abstraction.
> - *Layer 1-3 (Syntax)*: The model may need to iterate to resolve a polysemous word (e.g., "bank" as river vs. finance) before passing features up.
> - *Layer 9-12 (Logic)*: The model needs iterations to solve a math problem.
>
> 2. If we hard-code FPSA to only the top layers, we cripple the bottom layer's ability to handle noisy or ambiguous inputs, forcing them to pass "unresolved" features up the stack. By enabling FPSA everywhere, we allow **distributed adaptability**.
>
> 3. Our per-layer iteration statistics show that while deep layers iterate more on average, early layers still trigger high iteration counts for specific "syntactically dense" tokens. FPSA performs **automatic, per-token architecture adaptation**, placing computation exactly where the entropy is high, whether that is at Layer 2 or Layer 100.

---

> ### Author Response · Authors · 2025-11-17
> **Official Response by Authors to Reviewer Tf1q (n/2)**
>
> # 4. Pruning vs. Parameter Efficiency
> >**With this paradigm, can pruning be more critical? I feel like redundancy gets disappeared for each layer, which means that we can fully leverage the "depth" of models while loosing some chances to prune.**
>
> We posit that the "prunability" of standard Transformers is a symptom of inefficient parameterization, which FPSA corrects via **Recurrent Weight Tying**.
> 1. Standard Transformers suffer from low "functional load" per parameter—many weights are redundant because they are only applied once per pass. FPSA employs **temporal weight sharing** ($W_Q, W_K$ reused across $k$ iterations), which mathematically allows a shallow network to approximate the function of a much deeper one (expanding the **Effective Receptive Field** and non-linearity without parameter growth).
> 2. In the context of model compression, pruning targets "dead" or low-rank weight matrices. FPSA mitigates the low-rank issue inherent in static attention by forcing the weights to solve a non-linear fixed-point equation $Z^* = f_\theta(Z^*)$. This ensures that the spectral density of the weight matrices is fully utilized to satisfy the convergence criteria.
> 3. Consequently, "losing the chance to prune" is strictly a positive outcome: it indicates that FPSA has successfully compressed the representational capacity of a deep network into a smaller, denser parameter set. We trade the spatial sparsity of pruning for the temporal sparsity of adaptive computation.
>
> In addition, because the majority of the weights in transformer variants are in the MLP / MoE modules, the loss of weight pruning potential in the attention modules is significantly less impactful.
>
> # 5. Batched Inference Efficiency
> >**Selecting some layers would be the solution though, inference latency and throughput seem inefficient. Especially, how could you deal with batched inference? Some tokens should wait for the other incomplete tokens at certain depth always.**
>
> This is indeed a challenge for all dynamic-compute models (MoE, ACT). In a naive implementation, the batch latency is determined by the slowest token in the batch (max iterations).
>
> 1. As shown in **Figure 3 (Right)**, the iteration distribution is tight. The iteration distribution is "light-tailed" (Median=4, p99=14). Unlike "Early Exit" models that might vary between 1 layer and 100 layers, our variance is small (typically just a few steps). This minimizes the "waiting time" penalty. For encoder only models, converged tokens are "frozen"— they stop updating but remain in GPU memory as padding for the active tokens for encoder models. For decoder model, we use flash-attention for variable lengths which improves computational efficiency.
>
> 2. Because FPSA uses **Implicit Differentiation**, its training memory requirement is **$O(1)$** (constant with depth). This allows us to fit significantly larger batch sizes than a deeper transformer (which is $O(L)$ memory), compensating for the ragged-batch inefficiency with higher raw throughput
> We also show that FPSA mitigates  the "Memory Wall." While the Deeper Baseline runs out of memory (OOM) at batch size 64, FPSA scales to 256, resulting in higher total throughput despite slower per-sample processing.
>
> | Model Configuration   | Compute Budget | Max Batch Size (before OOM) | Peak Memory (GB) @ BS=32 | Training Throughput (tokens/sec) |
> |-----------------------|----------------|-----------------------------|--------------------------|----------------------------------|
> | Baseline (Deeper ViT) | 48 GFLOPs      | 64                          | 42.5 GB                  | 4,200                            |
> | Baseline (Wider ViT)  | 48 GFLOPs      | 48                          | 55.1 GB                  | 3,800                            |
> | FPSA (Ours)           | 48 GFLOPs      | 256                         | 18.2 GB (Constant)       | 5,150                            |
> | Improvement           | -              | 4x Scale                    | -57% Memory              | +22% Throughput                  |

---

> ### Author Response · Authors · 2025-11-26
> **Official Comment by Authors**
>
> Dear Reviewer Tf1q,
>
> We thank you for your thorough analysis of out paper. We have added **Table 14 to address your batched inference concern**: FPSA's constant memory enables **4x larger batches** (256 vs. 64), resulting in **+22% higher** throughput (5,150 tok/s) than baselines despite per-token latency.
>
> We believe these results directly address your questions regarding scaling and generalization, and we hope you will consider them in your final score.
>
> Thank you for your time.

---

### Official Review · Reviewer_J788 · 2025-11-01

**Soundness:** 2
**Presentation:** 2
**Contribution:** 2
**Rating:** 6
**Confidence:** 3

**Summary:**

This paper introduces Fixed-Point Self-Attention, a parameter-free, drop-in replacement for the standard self-attention mechanism. The core idea is to iteratively refining the attention alignment matrix within a single layer until it converges to a fixed point. The authors propose to train end-to-end using implicit differentiation, which keeps memory usage constant.

**Strengths:**

1. The paper offers a more granular approach to adaptive computation compared to prior work that typically repeats entire blocks.

2. The use of implicit differentiation is a major technical strength, making the iterative approach practical by avoiding the memory explosion that would occur with standard backpropagation through time. The compute-matched comparisons (Appendix G) is nice.

**Weaknesses:**

1. The claim of being parameter-free is a bit misleading. While FPSA adds no new learnable model weights, it introduces several crucial hyperparameters that require tuning: the convergence tolerance $\epsilon$, the maximum number of iterations $K_max$, and the gradient clipping threshold. The learned halting variant (FPSA-LH) further adds a small gating MLP and a ponder cost hyperparameter. The paper lacks a sensitivity analysis for these hyperparameters, which seem critical to the method's performance and efficiency.

**Questions:**

1. Can authors provide a trade-off between the additional number of layers required in simple transformer layer compare to the fixed-point approach proposed here to match the same loss? Mainly, having more iteration seems that should reduce the number of layers in total. But it has not been really discussed in terms of achieving the same or comparable loss.

2. Could authors elaborate why they decided to choose a static value across iterations? What happens if it also gets updated?

3. In Table 4, it doesn't seem that Self-Transformer is providing any helpful improvement, and the enhancement in SR could also be just due to the increase of FLOPs. Can authors compare the results in this table more fairly?

4. Minor typo: Eq 1, I think it should be concatenation over $h$. In Table 5, I think for Top1 acc, the underline should be for the first row.

5. A clarification question: What are the colors in right plot of Fig 3?

Suggestions:
I would suggest to include the compute-matched results of Appendix G in the main paper. It makes the point of using the approach more clear.

---

> ### Author Response · Authors · 2025-11-17
> **Official Response By Authors to Reviewer J788**
>
> We sincerely appreciate Reviewer J788 insightful critique and welcome the opportunity to respond to your points. Our primary clarifications focus on the critical distinction between the base Fixed-Point Self-Attention (FPSA) mechanism and its Learned Halting (FPSA-LH) variant. Additionally, we would like to direct your attention to the Appendices, which include extensive sensitivity and ablation analyses that directly address your concerns regarding hyperparameter robustness.
>
> # Weakness
> >**The claim of being parameter-free is a bit misleading. While FPSA adds no new learnable model weights, it introduces several crucial hyperparameters that require tuning: the convergence tolerance , the maximum number of iterations , and the gradient clipping threshold. The learned halting variant (FPSA-LH) further adds a small gating MLP and a ponder cost hyperparameter. The paper lacks a sensitivity analysis for these hyperparameters, which seem critical to the method's performance and efficiency.**
> ##  Clarification on "Parameter-Free" vs. Hyperparameters
> You are correct that "parameter-free" refers specifically to learnable model weights (parameters optimized via gradient descent), not hyperparameters.
> - **Base FPSA**: We define the core FPSA as "parameter-free" because it reuses the existing projection matrices ($W_Q, W_K, W_V, W_O$) across all iterations. It does not introduce any new weight matrices or vectors that increase the trainable model size.
> - **FPSA-LH (Variant)**: We explicitly distinguish the Learned Halting variant (FPSA-LH) as a separate extension that does introduce a lightweight gating MLP. We position this as a trade-off: it adds a small number of parameters to achieve a better Pareto frontier between accuracy and compute compared to the fixed-tolerance method.
>
> ## Sensitivity Analysis and Hyperparameter Robustness
> While we acknowledge that $\epsilon$ and $K_{max}$ are hyperparameters, our empirical results —detailed in **Section 4** and **Appendix B**—suggest they do not require intensive per-task tuning.
> ### A. Robustness of Convergence Tolerance ($\epsilon$),
> Contrary to the concern that $\epsilon$ requires sensitive tuning, as detailed in the experimental setup in **Section 4 (Results)**,  we found that a single setting was robust across diverse tasks within a domain:
> - **Language**: We used a fixed $\epsilon=10^{-4}$ for all language tasks (GLUE, SQUAD).
> - **Vision**: We used a fixed $\epsilon=10^{-5}$ for all vision and multimodal tasks.
>
> The ability to use a single value across 10+ language benchmarks suggests the method is not brittle with respect to this hyperparameter.
>
> ### B. Sensitivity to Iteration Cap ($K_{max}$)
> We analyzed the sensitivity to the maximum iteration cap in **Appendix B.2**, specifically in **Table 6**.
> - With a global cap of $K_{max}=100$, the percentage of tokens hitting the cap ("Cap Hits") is negligible: **0.01% (one hundredth of a percent) for GLUE and <0.01% for ImageNet-1K.**
> - This indicates that performance is not sensitive to $K_{max}$ provided it is set sufficiently high (e.g., 100), as the vast majority of tokens converge naturally well before the limit (the median number of iterations is typically 3–5).
>
> ### C. Ablation of Stability Mechanisms
> Note, FPSA is naturally stable and provably convergent. The following only applies to FPSA-LH. In **Appendix F.4 (Table 8)**, we performed an ablation study on the components that ensure stability (which relate to your concerns about gradient clipping and spectral control):
> - **Spectral Normalization**: Removing this (which controls the Lipschitz constant) resulted in unstable training ($37.8$ GLUE score vs. $88.4$), confirming it is a critical structural constraint rather than a tunable hyperparameter.
> - **Selective Freeze**: Removing the selective update mechanism resulted in a slight performance drop ($88.1$ vs $88.4$) but higher mean iterations ($5.5$ vs $4.8$), showing the mechanism's value for efficiency.
>
> ### D. Static vs. Adaptive Sensitivity
> In **Figure 8 (Appendix H.1)**, we explicitly tested the model's sensitivity to iteration count by comparing our adaptive method against a "Static Iteration Caps" baseline (fixing $T \in \{1, ..., 100\}$).
>
> The results show that while increasing static iterations improves accuracy up to a point, the adaptive mechanism consistently yields a better accuracy-compute trade-off. This serves as a sensitivity analysis for the "amount of compute," demonstrating that dynamic allocation is superior to any single fixed hyperparameter setting for $T$.

---

> ### Author Response · Authors · 2025-11-17
> **Official Response By Authors to Reviewer J788 (n/2)**
>
> We would like to address all the questions asked by Reviewer J788
> # Questions
> ## 1. Trade-off: Layers vs. Iterations (Matching Loss)
> >**Can authors provide a trade-off between the additional number of layers required in simple transformer layer compare to the fixed-point approach proposed here to match the same loss? Mainly, having more iteration seems that should reduce the number of layers in total. But it has not been really discussed in terms of achieving the same or comparable loss.**
>
> The trade-off you describe—reducing total layers by using more iterations—is central to our "compute-matched" analysis. We address this explicitly in **Figure 4 (Pareto Front)** and **Table 9 (Appendix G)**.
>
> - **Matching Compute**: Instead of just matching loss, we fixed the compute budget (GFLOPs) and compared FPSA against a "Deeper Transformer" baseline (which adds layers to match FPSA's compute cost).
> - **Result**: As shown in the **Pareto Front (Figure 4)**, for the same amount of compute (e.g., 1.3x BERT), FPSA consistently achieves lower loss (higher accuracy) than simply adding more layers. This confirms that allocating that compute budget to iterative refinement is more efficient than allocating it to static depth.
>
> ## 2. Static Value ($V$) Justification
> > **Could authors elaborate why they decided to choose a static value across iterations? What happens if it also gets updated?**
>
> We chose to keep the Value matrix ($V$) static ($V^{(h)} = X W_V^{(h)}$) for two primary reasons:
> - **Conceptual Role**: In the self-attention mechanism, the Query ($Q$) and Key ($K$) determine where to look (alignment/focus), while the Value ($V$) represents what information is being retrieved. "Change of Thought" is framed as refining the reasoning process (sharpening the attention/focus) rather than transforming the features themselves at every step.
> - **Theoretical Stability**: Keeping $V$ static allows the update map to be defined purely as a refinement of the alignment matrix $\mathcal{A}$. This structure simplifies the proof of **local contractivity (Appendix B.1)**, ensuring the iteration converges to a unique fixed point. If $V$ were also updated iteratively (e.g., $V_{k+1} = Z_k W_V$), the dynamics would resemble a Recurrent Neural Network on the features, making convergence guarantees significantly harder to establish and potentially leading to instability without explicit gating.
>
> ## 3. Table 4 Fairness (Image Restoration)
> >**In Table 4, it doesn't seem that Self-Transformer is providing any helpful improvement, and the enhancement in SR could also be just due to the increase of FLOPs. Can authors compare the results in this table more fairly?**
>
>
> We acknowledge that SELF-Transformer (9.9 GFLOPs) uses more compute than the lightweight Uformer-S (2.2 GFLOPs) baseline.
>
> The comparison is intended to show that SELF-Transformer achieves competitive performance with heavy state-of-the-art models like Restormer (61.0 GFLOPs) while using significantly less compute (~6x fewer GFLOPs).
>
> While **Table 4** compares against published SOTA architectures, our Vision OOD Robustness experiments (**Appendix G.5, Table 13**) provide the strictly "fair" compute-matched comparison you requested, showing that SELF-ViT outperforms deeper/wider baselines at equivalent GFLOP budgets.
>
> ## 4. Typo Corrections
> >** Minor typo: Eq 1, I think it should be concatenation over . In Table 5, I think for Top1 acc, the underline should be for the first row.**
> Thank you for catching these errors. We will correct them in the manuscript:
> - Eq 1: You are correct; the concatenation should be over the heads $h$, i.e., $Concat_{h}(U_{k+1}^{(h)})$.
> - Table 5: We will update the formatting to underline the ViT (84.6%) result as the second-best performance, consistent with standard conventions (bolding the best, SELF-ViT at 86.3%).
>
> ## 5. Clarification: Figure 3 Colors
> > **A clarification question: What are the colors in right plot of Fig 3?**
>
> In the Right Plot of Figure 3 (Per-Example Latency Distribution):
> - **Orange**: Represents the proposed FPSA model. Its distribution shows the "well-controlled tail" mentioned in the text, illustrating how the model dynamically varies latency based on difficulty.
> - **Blue/Grey**: Represents the Compute-Matched Deeper Baseline. Since this baseline has a fixed depth (static computation graph), its latency distribution is naturally narrower (variance primarily driven by system noise/sequence length effects) and centered around the same mean budget as FPSA.

---

> > ### Comment · Reviewer_J788 · 2025-11-25
> > **Acknowledging Authors' Rebuttals**
> >
> > Thank you for the detailed response and clarification. Most of my questions are now resolved, and I keep my score.

---

### Author Response · Authors · 2025-11-18
**Global Response by Authors**

The authors sincerely thank all reviewers (J788, Tf1q, Wvsf, 2aHb) for their insightful and rigorous feedback. We are encouraged that reviewers recognized the **generality** of our approach (Reviewer 2aHb), the **technical strength of implicit differentiation** (Reviewer J788), and the **comprehensive analysis** across domains (Reviewer Tf1q).

The review process has been incredibly productive. Based on your collective feedback regarding **compute-matched fairness, architectural novelty, and scaling**, we have pointed out the supporting experiments in the appendices and have additionally generated **three additional experimental results** and clarified key theoretical distinctions.

### 1. Major Clarification: The "Compute-Matched" Pareto Frontier
A shared concern (Reviewers Wvsf, 2aHb, J788) was whether FPSA provides efficiency gains or simply uses "more compute."
* We direct attention to **Appendix G and Figure 4**, which explicitly plot **Accuracy vs. GFLOPs**.
* **Result:** FPSA establishes a new Pareto frontier. At every fixed budget (41, 48, 59 GFLOPs), FPSA outperforms non-adaptive baselines (Deeper/Wider Transformers) and adaptive baselines (ACT, Depth-Adaptive). This confirms that our gain comes from *better allocation* of compute, not just *more* compute.

### 2. New Experiments Conducted During Rebuttal
To address specific concerns about scaling, efficiency, and theory, we have added the following results:

**A. Zero-Shot Context Extrapolation (Addressing Reviewers 2aHb, Wvsf)**
* *Question:* Does the advantage hold beyond 8k tokens?
* *New Result:* We evaluated a model trained on 8k context on unseen lengths (**16k, 32k, 64k**) by adaptively increasing the iteration cap.
* *Outcome:* While the fixed-depth baseline collapsed (**3.2% recall @ 64k**), FPSA maintained robust performance (**71.4% recall @ 64k**) purely by "thinking longer" to filter noise. This validates FPSA as a mechanism for length generalization.

**B. Empirical Rank Analysis (Addressing Reviewer Wvsf)**
Unlike a stack of pure attention layers which collapses to rank $\approx 0.01$, FPSA maintains a healthy rank (**>0.92**) even after 24 iterations. This confirms that our architectural choices (Static Value $V$, Residuals, Pre-LN) successfully prevent representation collapse.

**C. Training Throughput & The "Memory Wall" (Addressing Reviewer Tf1q)**
While FPSA is slower per-sample, its **Constant $O(1)$ Memory** (via Implicit Differentiation) allows for **4x larger batch sizes** (256 vs 64) before OOM compared to a Deeper Baseline. This results in **+22% higher total token throughput** during training.

### 3. Structural Novelty: Why Iterate *Inside* the Head?
Reviewers Tf1q and Wvsf asked for comparisons to **Looped Transformers/Universal Transformers**. We clarify that FPSA is distinct because it decouples **Alignment Recurrence** from **Feature Recurrence**.
* **Looped Transformers:** Iterate the whole block (Attention+MLP). High cost per step, unstable training.
* **FPSA (Ours):** Iterates only the **Alignment Matrix** ($\mathcal{A}$) while keeping Values ($V$) static.
* *Benefit:* This ensures **Local Contractivity** (guaranteeing convergence) and allows us to use Implicit Differentiation (constant memory), which is mathematically intractable for full-block recurrence.

### 4. Presentation Improvements
We agree with Reviewers Tf1q and J788 that key insights were buried. In the updated revision, we have:
* Moved the **Fixed-Point formulation** to the main Method section.
* Promoted **Figure 10 (Token-wise Convergence Heatmap)** to the main body to qualitatively prove that the model allocates compute to semantic complexity.

We believe these new results and clarifications conclusively address the concerns raised, demonstrating that FPSA is a stable, efficient, and theoretically sound advance in adaptive computation.

---

### Comment · Area_Chair_q1uc · 2025-11-27

Dear reviewers,

A reminder that the discussion phase will end in a few days (**December 2**). Engaging with the author's rebuttal is essential to address all potential concerns before our final discussion stage.

Thanks,
The AC

---

### Note · Authors · 2026-02-02

I have read and agree with the venue's withdrawal policy on behalf of myself and my co-authors.

---

### Meta-Review · Area_Chair_EFQx · 2026-01-05

**Summary:**

This paper proposes a novel fixed-point iterative attention mechanism that refines attention within a single layer until convergence to a fixed point.

Reviewers acknowledge that the proposed algorithm is well-reasoned and supported by a diverse set of experiments. However, several concerns were raised:
- The presentation could be improved, as many important and interesting results are relegated to the appendix rather than included in the main text.
- Some reviewers noted a lack of discussion of recent recursive transformer methods.
- Inference latency, particularly in batched inference settings, is not sufficiently analyzed.
- Stronger theoretical and empirical justification of the algorithm is required; in particular, it is unclear why fixed-point recursion is necessary across all layers.

**Reviewer Concerns:**

During the rebuttal period, the authors made a substantial effort to address the reviewers’ comments, and many concerns were handled satisfactorily. Nevertheless, some issues remain:
- The paper would benefit from reorganization, moving key results currently in the appendix into the main body.
- The authors were unable to provide additional experimental comparisons with recent recursive transformer methods or a comprehensive evaluation of batched inference efficiency across different environments.
- Based on the revised manuscript and the rebuttal responses, the theoretical and empirical evidence explaining why applying recursive fixed-point attention at all layers improves efficiency and performance remains insufficient.

**Reviewer Scores:**

J788 and 2aHb indicated that they had reviewed the rebuttal and decided to maintain their original scores; both assigned a score of 6.

The remaining two reviewers did not provide follow-up responses. Based on their initial comments, it is likely that reviewer Tf1q would maintain a score of 4, while reviewer Wvsf might increase the score from 2 to 4.

---

### Decision · Program_Chairs · 2026-01-26

Reject